



# How Asian aerosols impact regional surface temperatures across the globe

Joonas Merikanto[1], Kalle Nordling[1], Petri Räisänen[1], Jouni Räisänen[2], Declan O'Donnell[1], Antti-Ilari Partanen[1], Hannele Korhonen[1]

[1]Finnish Meteorological Institute, Helsinki, Finland
[2]INAR/Physics, University of Helsinki, Helsinki, Finland

*Correspondence to*: Joonas Merikanto (joonas.merikanto@fmi.fi)

**Abstract.** South and East Asian anthropogenic aerosols mostly reside in an air mass extending from the Indian Ocean to the North Pacific. Yet the surface temperature effects of Asian aerosols spread across the whole globe. Here, we remove Asian anthropogenic aerosols from two independent climate models (ECHAM6.1 and NorESM1) using the same representation of aerosols via MACv2-SP (a simple plume implementation of the 2nd version of the Max Planck Institute Aerosol Climatology). We then robustly decompose the global distribution of surface temperature responses into contributions from atmospheric energy flux changes. We find that the horizontal atmospheric energy transport strongly moderates the surface temperature response over the regions where Asian aerosols reside. Atmospheric energy transport and changes in clear-sky longwave radiation redistribute the temperature effects efficiently across the Northern hemisphere, and to a lesser extent also over the Southern hemisphere. The model-mean global surface temperature response to Asian anthropogenic aerosol removal is 0.26±0.04 °C (0.22±0.03 for ECHAM6.1 and 0.30±0.03 °C for NorESM1) of warming. Model-to-model differences in global surface temperature response mainly arise from differences in longwave cloud (0.01±0.01 for ECHAM6.1 and 0.05±0.01 °C for NorESM1) and shortwave cloud (0.03±0.03 for ECHAM6.1 and 0.07±0.02 °C for NorESM1) responses. The differences in cloud responses between the models also dominate the differences in regional temperature responses. In both models, the Northern hemispheric surface warming amplifies towards the Arctic, where the total temperature response is highly seasonal and weakest during the Arctic summer. We estimate that under a strong Asian aerosol mitigation policy tied with strong climate mitigation (Shared Socioeconomic Pathway 1-1.9) the Asian aerosol reductions can add around 8 years' worth of current day global warming during the next few decades.

## 1 Introduction

Understanding how regional climates respond to different climate forcers is crucial for assessing how climate change impacts societies. Samset et al. (2018) showed that anthropogenic aerosols cool the global mean surface temperature in four latest generation climate models by between 0.5K – 1.1K. However, the regional impacts of anthropogenic aerosols on surface temperatures remain particularly complicated to unravel (Persad and Caldeira, 2018; Nordling et al., 2019).





Due to the short lifetime of aerosols, their distribution in the atmosphere is highly heterogeneous and dependent on the location of their emissions and on various dynamical and microphysical processes influencing also their properties and climate effects. Aerosols give rise to both local and remote temperature responses, so that the geographic distributions of aerosol radiative
forcing and temperature effects are largely dislocated (Shindell et al., 2010; Nordling at al., 2019). Furthermore, same aerosol emissions originating from different regions vary in their climate forcing efficacies, with their global surface temperature response per unit global radiative forcing differing by factors between 2 to 14, depending on aerosols species and the models used (Kasoar et al., 2018; Westerveld, et al., 2020; Persad and Caldeira 2018).

During the past decades air pollution levels in Europe and North America have decreased considerably while grown in South and East Asia. These opposing changes in air pollution have kept the overall global anthropogenic aerosol radiative forcing close to constant since the mid-1970s (Murphy 2013; Fiedler et al., 2018). South and East Asia have become the dominant sources of anthropogenic aerosol emissions (Lamarque, 2010). Consequently, air pollution has become a major health problem in Asia. Ambient aerosol pollution reduces the life-expectancy by 1.24 years in East Asia and by 1.56 years in South Asia
(Apte et al., 2018), and is attributable to 0.67 million deaths per year in India alone (Balakhrisnan et al., 2019). Shared Socioeconomic Pathways (SSPs) predict that strong air pollution mitigation policies (SSP1-1.9) could reduce the Asian aerosol emissions from their 2015 levels by 55% already by 2030 and by 90% by 2100 (Lund et al., 2019, Fig S2).

Large past changes and potentially at least equally significant future changes in Asian aerosols have prompted recent studies
on their global and regional surface temperature effects. Kasoar et al. (2016) used three climate models (HadGEM3-GA4, CESM1, and GISS-E2) to study regional surface temperature responses to the removal of $SO_2$ emissions from China. Two of the three models showed Northern hemispheric warming due to aerosol removal but of significantly different magnitude, while the third model showed no significant surface temperature responses. The authors pinpointed the mixed results from the models to their different treatments of aerosol microphysical processes and aerosol-cloud interactions. Westerweld et al. (2020) also
used three climate models (GFDL, CESM1, and GISS-E2) to investigate the surface temperature responses to the removals (or significant reductions) of aerosol sources from several different regions, including China and India. Overall, the models varied in aerosol radiative forcings and regional temperature response patterns associated with Asian aerosol reductions, but suggested that the reductions mostly result in significant surface temperature increase across the Northern hemisphere, and particularly over the Arctic. Persad and Caldeira (2018) used the CAM5 model to place an equivalent to China's total annual
year 2000 anthropogenic aerosol emissions at different locations around the globe. They found that emissions placed in China cooled the whole Northern hemisphere, while the same emissions placed in India resulted in a mixed regional response of warming and cooling.

Here, we explore the global and regional surface temperature responses to a complete removal of South and East Asian aerosols
using two different climate models, ECHAM6.1 and NorESM1. As in Nordling et al. (2019), we use an identical description
of anthropogenic aerosols in both models. The use of identical aerosols across the models allows us to study the similarities
and differences in model dynamical responses to aerosols, and exclude the model response perturbations that result from
differences in modeled aerosols. Further, we aim to understand the robustness of changes in the climate system that lead to the
local and remote changes in surface temperatures.


It is complicated to resolve the pathway from a climate forcing to a regional surface temperature response in climate models
even for globally homogeneous greenhouse gases, let alone for aerosols. A significant climate perturbation results in a complex
set of responses in general circulation patterns, cloud properties, surface albedo, atmospheric water vapor concentrations et
cetera. Surface temperature responses result from a combination of all these different climate feedbacks. Therefore, even a
seemingly robust regional surface temperature signal in different climate models may result from a different combination of
feedbacks that sums up to a similar temperature response.

Räisänen (2017) presented a new method built around the concept of effective planetary emissivity for a robust decomposition
of the energetic components that sum up to the geographic distribution of surface temperature responses. Here, we extend the
method to better resolve the longwave cloud feedback using radiative kernels, and apply it for the analysis of the model results.
The method allows separating the contributions from atmospheric heat transport, changes in shortwave and longwave radiation
related to clear sky and clouds, surface energy fluxes, and surface albedo to a local surface temperature response.

## 2 Method

### 2.1 Model experiments and analysis

We use ECHAM6.1 (Stevens et al., 2013) and NorESM1 (Bentsen et al., 2013, Iversen et al. 2013; Kirkevåg et al. 2013)
general circulation models to carry out 100-year slab ocean equilibrium runs for the present day (year 2005) atmosphere
without South and East Asian anthropogenic aerosols, but leaving all other aerosol sources intact. The last 60 years of
equilibrated climate data from each simulation are used for the analysis. These runs are compared to otherwise identical
baseline climate runs of same length but having all aerosol sources on. For the baseline, we use the same ECHAM6.1 and
NorESM1 model runs as presented in Nordling et al., (2019).
The background aerosols for ECHAM6.1 are prescribed using the climatology of Kinne et al. (2013), while for NorESM1,
they are simulated by the model's bottom-up aerosol microphysics scheme (Kirkevåg et al. 2013). In both cases, the impact of
modern-day (year 2005) anthropogenic aerosols is represented via the MACv2-SP climatology (Stevens et al. 2017). MACv2-
SP uses in-situ observations of aerosol optical depth (AOD) for the top-to-bottom representation of aerosol-radiation effects,
and calculates the aerosol direct and first indirect effects through changes in fixed three dimensional AOD fields with monthly



time resolution. The anthropogenic impact on AOD is represented through nine different aerosol plumes, which together represent the sources and transport of anthropogenic aerosols, including biomass burning. In the runs without Asian anthropogenic aerosols we have turned off plume numbers 3 and 4. The direct and indirect instantaneous aerosol radiative are calculated online in the models using double radiation calls. The global instantaneous forcing can be modelled near-identically

with MACv2-SP in ECHAM6.1 and NorESM1, albeit there are some model-to-model differences related to model –specific representations of clouds, surface albedo, and background aerosol (Nordling et al., 2019). The effective radiative forcing (analyzed by Fiedler et al. (2019) for multi-decadal fixed-SST runs for ECHAM6.3 and NoreSM1 models using the same background aerosol representations as here) shows somewhat larger model-to-model variations, but the geographic patterns of the effective radiative forcings with MACv2-SP are close to those of instantaneous radiative forcings. The differences in global

anthropogenic aerosol radiative forcings between the ECHAM6.1 and NorESM1 models with MACv2-SP aerosols are small enough to be insignificant for the obtained temperature responses, as discussed in Nordling et al. (2019). However, different representations of natural background aerosol in the models can lead to differences in obtained indirect aerosol forcing (Carslaw et al., 2013; Fiedler et al., 2019), and this is the case also for anthropogenic Asian aerosols when using the model-intrinsic background aerosol representations in NorESM1 and ECHAM6.1, as we will discuss later. Here, both models were

coupled to their intrinsic mixed-layer (slab) ocean model representations (for ECHAM6.1 see Roekner et al., 2003; for NorESM1 see Bitz et al., 2012), and hence changes in ocean currents are not accounted for in our analysis.

The analysis of results is based on monthly-mean values of data, calculated separately for each month in the 60 -year timeseries. The response uncertainties in global-mean values are calculated as standard error of means using a 95% confidence interval

for individual models, and as a pooled standard error of mean with a 95% confidence interval for responses averaged over the two models. The statistical significance of regional responses is evaluated using a Student's t-test with an autocorrelation correction according to Zwiers and von Storch (1995).

## 2.2 Temperature response decomposition

We decompose the distribution of local surface temperature responses to local changes in atmospheric energetic components

using a method presented in Räisänen (2017). The method only requires standardly archived climate model output for the decomposition.

The rate of change of total energy in an atmospheric column is

$\frac{\delta E}{\delta t} = SW^{\downarrow}_{TOA} - LW^{\uparrow}_{TOA} - F^{\downarrow}_{SURF} + C^{\rightarrow}$,     (1)





where $SW_{TOA}^{\downarrow}$ is the net incoming shortwave radiation and $LW_{TOA}^{\uparrow}$ is the outgoing longwave radiation at the top of the atmosphere, $C^{\rightarrow}$ is the net horizontal heat transport into the column (energy flux convergence), and the net downward heat flux into the surface is given by


$$F_{SURF}^{\downarrow} = SW_{SURF}^{\downarrow} + LW_{SURF}^{\downarrow} + SH + LH, \tag{2}$$

where $SW_{SURF}^{\downarrow}$ and $LW_{SURF}^{\downarrow}$ are the net shortwave and longwave radiation fluxes into the surface, and $SH$ and $LH$ are the sensible and latent heat fluxes, respectively. To relate Eq. (1) with the surface air temperature $T$ one defines (Räisänen and

Ylhäisi, 2015; Räisänen, 2017)

$$LW_{TOA}^{\uparrow} = \varepsilon_{eff}\sigma T^4, \tag{3}$$

where the effective planetary emissivity $\varepsilon_{eff}$ is essentially a measure of the local atmospheric greenhouse effect. Substituting

Eq. (3) into Eq. (1) gives

$$\varepsilon_{eff}\sigma T^4 = SW_{TOA}^{\downarrow} - F_{SURF}^{\downarrow} + C^{\rightarrow} - \frac{\delta E}{\delta t} \tag{4}$$

Then, letting [] to mark the mean state between baseline and perturbed climates, the change in Eq. (4) between the two climate

states can be written as

$$\sigma[\varepsilon_{eff}]\Delta(T^4) = -\sigma\Delta\varepsilon_{eff}[T^4] + \Delta SW_{TOA}^{\downarrow} - \Delta F_{SURF}^{\downarrow} + \Delta\left(C^{\rightarrow} - \frac{\delta E}{\delta t}\right) \tag{5}$$

Linearizing the left hand side of Eq. (5) as


$$\sigma[\varepsilon_{eff}]\Delta(T^4) \approx 4\sigma[\varepsilon_{eff}][T^3]\Delta T = D\Delta T \tag{6}$$

allows the decomposition of surface temperature response into changes in energy flux components in the atmospheric column,

$\quad \Delta T = \Delta T_{LW} + \Delta T_{SW} + \Delta T_{SURF} + \Delta T_{CONV}, \tag{7}$

where





$$\Delta T_{LW} = -\frac{\sigma \Delta \varepsilon_{eff}[T^4]}{D},$$ (8a)


$$\Delta T_{SW} = \frac{\Delta SW_{TOA}^\downarrow}{D},$$ (8b)

$$\Delta T_{SURF} = -\frac{\Delta F_{SURF}^\downarrow}{D},$$ (8c)

$$\Delta T_{CONV} = \frac{\Delta\left(C^\rightarrow - \frac{\delta E}{\delta t}\right)}{D}.$$ (8d)

We mark the surface temperature change due to horizontal heat transport and the change in the energy storage (Eq. 8d) collectively as CONV. Annually, the change in the energy storage of an atmospheric column averages to zero in an equilibrium climate (Porter et al., 2010; Räisänen, 2017), and Eq. (8d) corresponds to the difference in horizontal heat transport between
two equilibrium climates. However, on monthly and seasonal timescales the changes in atmospheric energy storage can be significant.

The terms on the right hand side of Eqs. (8a-b) can be further expanded to separate the surface temperature responses due to clear-sky and cloud radiative effects. The standard climate model output contains radiative fluxes both for all-sky and clear-
sky (CS) conditions, so that the temperature response to longwave cloud radiative effect can be obtained as

$$\Delta T_{LW_{CRE}} = \Delta T_{LW} - \Delta T_{LW_{CS}} = -\frac{\sigma \Delta \varepsilon_{eff}[T^4]}{D} + \frac{\sigma \Delta \varepsilon_{eff,cs}[T^4]}{D}$$ (9)

Räisänen (2017) calculated the surface temperature response due to changes in longwave cloud emissivity as $\Delta T_{LW_{CRE}}$, but
noted that it is a negatively biased approximation of the actual cloud long-wave feedback, as also discussed by Soden et al., (2004). Here, we extend the calculation to allow for a more precise separation of thermal radiation to its clear-sky and cloud contributions with the help of radiative kernels. Radiative kernels are climate model –derived radiative responses to small changes in climate state, such as to changes in atmospheric and surface temperature water vapor under clear-sky and all-sky conditions. We use the radiative kernels of Block and Mauritsen (2013) and their Eq. (4) to calculate a corrected longwave
cloud feedback $\Delta LW_{cld} \approx \Delta LW_{CRE} - \Delta LW_{cor}$, namely

$$\Delta LW_{cor} = (K_T - K_T^{clr})\Delta T + \sum_i \left(K_{T_i} - K_{T_i}^{clr}\right)\Delta T_i + \sum_i \left(K_{w_i} - K_{w_i}^{clr}\right)\Delta(\ln q)_i,$$ (10)





Where $K_T$ and $K_w$ are different model level mass-weighted radiative kernels, $q$ is the specific humidity, and the summations are carried over the model levels $i$. Block and Mauritsen (2013) generated their radiative kernels with the ECHAM6 climate model, and here we apply the kernels both to the ECHAM6.1 and NorESM1 models. This should bring no major bias for the NorESM1 calculations, as Myhre et al. (2018) showed that radiative kernels do not significantly depend on the specific model used for their construction. The calculated correction is used to redistribute the effect of $\Delta\varepsilon_{eff}$ between the cloud and clear-sky terms as

$$\Delta T_{LW_{cld}} = \Delta T_{LW_{CRE}} - \frac{\Delta LW_{cor}}{D} \tag{11a}$$

$$\Delta T_{LW_{clr}} = \Delta T_{LW_{CS}} + \frac{\Delta LW_{cor}}{D} \tag{11b}$$

where $\Delta T_{LW_{cld}}$ and $\Delta T_{LW_{clr}}$ are the corrected longwave cloud and clear-sky temperature responses.

As discussed in Räisänen (2017), the surface and top-of-atmosphere shortwave radiative responses for clear-sky and all-sky conditions can also be further separated to physically more meaningful terms using the Approximative Partial Radiative Perturbation (APRP) method of Taylor et al. (2007).

$$\Delta T_{SW} = \Delta T_{SW_{IN}} + \Delta T_{SW_{clr}} + \Delta T_{SW_{cld}} + \Delta T_{SW_{Albedo}} + \Delta T_{SW_{NL}}, \tag{12}$$

where $\Delta T_{SW_{IN}}$ corresponds to changes in incoming solar radiation (zero in our model experiments), $\Delta T_{SW_{clr}}$ is the corrected clear-sky shortwave temperature response, $\Delta T_{SW_{cld}}$ is the shortwave cloud response, $\Delta T_{SW_{Albedo}}$ the temperature response due to changes in surface albedo, and $\Delta T_{SW_{NL}}$ is a non-linear correction term, small enough to be insignificant for the analysis.

Hereafter, we use the subscripts in $\Delta T$ terms as shorthand notations when discussing the various temperature responses (so that $\Delta T_{SW_{clr}}$ is discussed as $SW_{clr}$ etc.).

## 3 Results

### 3.1 Radiative forcing

Figure 1 shows the net change in instantaneous top-of-atmosphere aerosol radiative forcing, $\Delta IRF$, due to removal of South and East Asian anthropogenic aerosols, calculated as an average over the full 60-year equilibrated climate data sets over both models as





$\Delta IRF(removed\ S\&E\ Asian\ aerosols) = IRF(no\ S\&E\ Asian\ aerosols) - IRF(all\ aerosols).$    (13)

Note that since we here remove the Asian aerosols from the models, $\Delta IRF$ is positive in sign, i.e. that of warming. $\Delta IRF$ further breaks into $\Delta IRF = \Delta IRF_d + \Delta IRF_{id}$, where $\Delta IRF_d$ describes the change in aerosol direct radiative forcing due to the net change in direct radiation attenuation of aerosols through their scattering and absorption of solar radiation. $\Delta IRF_{id}$ is the change
in indirect radiative forcing (the Twomey effect) between the runs without and with South and East Asian aerosols. The geographical pattern of $\Delta IRF$ is nearly identical for ECHAM6.1 and NorESM1, with the model-to-model correlation coefficient of 0.99. However, the modeled globally averaged $\Delta IRF$ differs slightly between the models, being $0.38\pm0.00$ Wm$^{-2}$ for ECHAM6.1 and $0.41\pm0.00$ Wm$^{-2}$ for NorESM1, with a model mean of $0.40\pm0.00$ Wm$^{-2}$. Results for individual models are shown in the Appendix Fig. A1.

In the models, $\Delta IRF$ due to removal of Asian aerosols is concentrated on a distinctive patch over the region surrounding the aerosol sources. The change in local radiative forcing reaches up to 8.3 Wm$^{-2}$ over SE China. The change in direct radiative forcing $\Delta IRF_d$ in the models is responsible for slightly over a half ($0.22\pm0.00$ Wm$^{-2}$; $0.23\pm0.00$ Wm$^{-2}$ for ECHAM6.1 and $0.20\pm0.00$ Wm$^{-2}$ for NorESM1 with a model-to-model correlation coefficient 0.96) of the total globally averaged $\Delta IRF$, and
more focused on the polluted regions than the change in indirect forcing $\Delta IRF_{id}$ ($0.18\pm0.00$ Wm$^{-2}$; $0.15\pm0.00$ Wm$^{-2}$ for ECHAM6.1 and $0.21\pm0.00$ Wm$^{-2}$ for NorESM1 with a model-to-model correlation coefficient 0.94), which spreads more evenly over a larger area. The higher model-to-model correlation coefficient for $\Delta IRF$ than for $\Delta IRF_d$ and $\Delta IRF_{id}$ separately indicates a cancellation of regional model-to-model differences when changes in direct and indirect radiative forcings are summed up. This cancellation of differences in $\Delta IRF$ suggests that differences in modelled cloud fields mainly distribute $\Delta IRF$
differently to its $\Delta IRF_d$ and $\Delta IRF_{id}$ components in the models, while differences in modelled background aerosols likely also play a role in model-to-model difference in $\Delta IRF_{id}$.

### 3.2 Annually averaged temperature response decomposition

We first describe the commonalities in modeled surface temperature responses to the omission of S&E Asian aerosols in the two models, before discussing their differences. The average global equilibrium temperature response $\Delta T$ to the removal of
aerosols across the two models is shown in Fig. 2A (individual models shown in Figs. A2A and A3A, and global-level results are collected on Table A1). The regional distribution of surface temperature response is strikingly different from the distribution of S&E Asian aerosol forcing, with surface warming spreading over the entire Northern Hemisphere, and to a lesser extent also to the Southern hemisphere. Indeed, significant warming extends to regions where no change in aerosols is modeled, such as over to the North American continent (0.5 K) and to Arctic regions with warming exceeding 1 K. The
warming over the region with the strongest change in local aerosol forcing (SE China) is 1.5 K with a local climate sensitivity



of 0.18 K/(Wm$^{-2}$) (i.e., 1.5 K / 8.3 Wm$^{-2}$), while the globally averaged warming is 0.26±0.04 K with a climate sensitivity of 0.65±0.11 K/(Wm$^{-2}$).

As described in Methods, the total local temperature response can be represented as a sum of responses in clear and cloudy-sky shortwave (SW$_{clr}$ and SW$_{cld}$) and longwave (LW$_{clr}$ and LW$_{cld}$) radiation, surface albedo (ALBEDO), surface energy fluxes (SURF), and an energy convergence term (CONV) representing the horizontal transport of heat for annually averaged results. The annually averaged temperature responses for each of the energetic terms, averaged over the 60 year sets of equilibrated climate runs with both models, are shown in Fig. 2B-H. The sum of surface temperature responses to individual energetic terms (sum over panels B-H) reproduces the total surface temperature response in panel A with a spatial correlation coefficient 260 cc=0.998 and an identical global mean.

First it can be noted that the geographical distribution of temperature responses due to changes in clear-sky shortwave radiation, SW$_{clr}$ (Fig. 2B) resembles closely the distribution of shortwave direct radiative forcing, $\Delta IRF_d$ (Fig. 1B), with a correlation coefficient cc=0.96. SW$_{clr}$ is also one of the major energetic terms of the total globally averaged temperature response, 265 responsible for 0.08±0.01 K of the total globally averaged temperature response of 0.26±0.04 K.

Over the whole region of positive radiative forcing ($\Delta IRF$ in Fig. 1A) the warming is moderated by the cooling caused by the transport of energy away from the region, CONV (Fig. 2C). CONV also efficiently redistributes the temperature effects across the globe. Since CONV only acts by horizontally redistributing atmospheric energy, its effect on the global surface temperature 270 response is effectively zero (-0.01±0.02 K).

Unlike for the connection between SW$_{clr}$ and $\Delta IRF_d$, the geographical distribution of temperature responses due to changes in cloudy-sky shortwave radiation, SW$_{cld}$ (Fig. 2D) corresponds only weakly to the geographical distribution of the change in shortwave cloud radiative forcing $\Delta IRF_{id}$ (Fig. 1C) (cc=0.23). Indeed, while there is pronounced positive $\Delta IRF_{id}$ in South 275 Asia and western subtropical North Pacific in Fig. 1C, much of the warming response in this region appears actually in the LW$_{cld}$ term (Fig. 2E). This is because of feedbacks that lead to changes in cloud cover and other cloud properties. Clouds both reflect SW radiation and reduce outgoing longwave radiation (e.g., Loeb et al. 2018), and changes in cloud amount tend to have opposing effects on SW$_{cld}$ and LW$_{cld}$. The average total cloud cover change in the models is shown in Fig. 3A. The global distribution of cloud cover changes correlates strongly with LW$_{cld}$ (cc=0.77) and anti-correlates with SW$_{cld}$ (cc= -0.74). Only 280 by summing SW$_{cld}$ and LW$_{cld}$ (Fig 2I) one can again recognize the warming response to $\Delta IRF_{id}$ (Fig 1C) (cc=0.70). There is a marked and statistically significant increase in cloud cover over India, the Indochinese peninsula, and western subtropical North Pacific, accompanied with a strong decrease in SW$_{cld}$ and increase in LW$_{cld}$. The strong increase in cloud cover over India and the Indochinese peninsula leads to a weaker overall surface temperature response (Fig. 2A) in these regions. In contrast, the decrease of cloud cover over East Asia amplifies the temperature response over the region. Further, the changes





in clouds also contribute to remote temperature responses, such as to a weakening of the cloud cover over Mediterranean and central Asia with compensating surface temperature effects from the $SW_{cld}$ and $LW_{cld}$ pathways. Overall, the combined effect of clouds ($SW_{cld}$ + $LW_{cld}$) on the globally averaged temperature response is 0.08±0.04 K.

Together with the horizontal energy transport CONV, also the clear-sky longwave response $LW_{clr}$ acts as a strong redistributor
of the surface temperature changes across the globe. Similarly to $CO_2$ forcing (Räisänen, 2017), $LW_{clr}$ (0.08±0.03 K) is one of the major terms in the overall global temperature response also for aerosols. This is somewhat counterintuitive, as the modeled aerosols only impact the shortwave radiation in clear and cloudy skies. The geographical distribution of $LW_{clr}$ mainly results from a combination of atmospheric water vapor and lapse rate feedbacks (Pithan and Mauritsen, 2014; Räisänen, 2017), but the separation of these feedbacks is not pursued in this study.


Fig. 2F shows the annual average change across both models in ALBEDO, that is the surface temperature response to the change in surface albedo. The change in surface albedo is related to changes in snow and sea-ice cover, but interestingly the surface albedo (ratio between reflected and incoming surface shortwave radiation) also changes over India in both models, likely due to changes in the ratio of direct vs. diffuse solar radiation. The surface albedo change further pushes the geographical
distribution of warming towards Northern latitudes. The globally averaged temperature effect of the surface albedo change is nevertheless small (0.02±0.01 K)

The annually averaged temperature response due to changes in surface energy flux, SURF (Fig. 2G), is zero over the continents as there is no net annual exchange of heat between continental surface and the atmosphere regardless of climate state, and
nonzero mainly over ocean regions where there are changes in sea ice cover. In climate runs with fully coupled ocean models, instead of slab-ocean models used here, the annually averaged oceanic surface terms could be larger due to changes in oceanic circulation and heat transport. Since we have run the modeled climates to an equilibrium, the yearly averaged global SURF is zero (0.00±0.04 K); yet, it introduces a significant noise term in the results. However, oceanic SURF term plays an important role in the seasonal cycle of regional temperature responses, as we will discuss further when describing the seasonality of
modelled temperature responses.

### 3.3 Model-to-model differences in regional temperature responses

The parenthesis in Figs. 1, 2 and 3 show the model-to-model correlation coefficients for the geographical distributions of changes in radiative forcings, temperature response terms and cloud cover due to the removal of South and East Asian anthropogenic aerosols. The coefficients describe the geographical uniformity of responses for the ECHAM6.1 and NorESM1
climate models using the same representation of anthropogenic aerosols via the MACv2-SP aerosol scheme.



The strong correlation between modelled change in aerosol direct radiative forcing $\Delta IRF_d$(cc=0.96) translates into a strong correlation between the modelled surface temperature response due to SW$_{clr}$ (cc=0.91). However, the strong correlation for the change in aerosol indirect radiative forcing $\Delta IRF_{id}$(cc=0.96) does not result in a strong correlation between the modelled SW$_{cld}$

(cc=0.28). This is due to different changes in modelled cloud fields, and the high sensitivity of SW$_{cld}$ to such changes. As discussed in the previous section, the changes in cloud fields also lead to changes in LW$_{cld}$. In both models the change in LW$_{cld}$ (cc=0.46) is particularly pronounced over the Asian monsoon region, where the cloud cover increases due to the omission of S&E Asian anthropogenic aerosols. The total surface temperature response due to clouds, SW$_{cld}$+ LW$_{cld}$ (cc=0.37) has a similar correlation as the change in total cloud cover (cc=0.37) between the models. The distribution of annual average surface

temperature responses due to changes in atmospheric energy transport, CONV (cc=0.64), is modeled relatively robustly across the models, given that CONV extends over both hemispheres. The correlation between annual LW$_{clr}$ terms (cc=0.52) is modest, and differences in LW$_{clr}$ contribute to model differences in the total temperature response particularly over North Asia. The surface temperature responses to albedo changes in the models, ALBEDO (cc=0.23), have a rather weak correlation, but much of the deviation in ALBEDO responses results from sporadic differences in modelled Southern Ocean sea ice.


The total surface temperature response $\Delta T$(cc=0.65) due to removal of S&E Asian anthropogenic aerosols using the MACv2-SP aerosol scheme has a weaker correlation than the temperature response due to removal of all anthropogenic MACv2-SP aerosols in the same models (cc=0.78) (Nordling et al., 2019). The total globally and annually averaged surface temperature responses in the models due to the removal of S&E Asian anthropogenic aerosols (0.22±0.03 KK for ECHAM6.1 and

0.30±0.03 K for NorESM1) also differ more than the corresponding values for the complete removal of anthropogenic aerosols (0.48±0.04 K for ECHAM6.1 and 0.50±0.06 K for NorESM1, with error given here using a 95% confidence interval). For the removal of S&E Asian anthropogenic aerosols modelled here, the largest contributors to the differences in modeled globally and annually averaged surface temperature responses between the two models are the cloud terms SW$_{cld}$ (0.03±0.03 K for ECHAM6.1 and 0.07±0.02 K for NorESM1) and LW$_{cld}$ (0.01±0.01 K for ECHAM6.1 and 0.05±0.01 K for NorESM1).

**3.4 Seasonal cycle of temperature responses across NH latitudes**

The seasonal cycle of latitudinal temperature responses is shown in Fig. 4A for ECHAM6.1 and in Fig. 4B for NorESM1. The figures also highlight the latitudinal dislocation of the change in aerosol radiative forcing and the resulting temperature response. In both models the change in radiative forcing peaks between 20-30 °N, but the temperature responses are strongest over the High North.


The seasonality of the latitudinal surface temperature responses (star symbols in Fig. 4) is modest in both models from low to mid-latitudes, with opposing changes in energetic terms contributing to balancing cooling and warming seasonal responses. Throughout the Northern hemisphere both shortwave and longwave clear-sky terms (SW$_{clr}$ and LW$_{clr}$ shown with color bars) remain positive during all seasons. Surface temperature changes due to cloud shortwave responses (SW$_{cld}$) are strongest during

the summer, being positive in the mid-latitudes but mostly negative elsewhere. The cloud longwave term (LW$_{cld}$) typically opposes the SW$_{cld}$ responses, and particularly strongly over the 10-20 °N band due to opposing responses to changes in cloudiness in the region. The change in the net oceanic surface energy flux, (SURF), amplifies the summer warming in 0-20 °N during the Asian monsoon, and overall the changes in oceanic surface fluxes tend to regulate the modest seasonality of temperature responses from low to mid-latitudes, and amplify the seasonality of the response in the Arctic. Atmospheric

energy transport and storage (CONV) also regulates the modest seasonality of responses from low to mid-latitudes together with SURF, although these terms tend to oppose each other. The seasonal CONV terms grow from mostly negative at low northern latitudes towards mostly positive at high northern latitudes, reflecting the increase in atmospheric energy transport towards High North.

The differences in modelled latitudinal temperature responses become larger from 50 °N northwards, where the direct influence from the change in aerosol radiative forcing diminishes. Between 50-70 °N warming from the longwave clear-sky term is stronger in NorESM1 than in ECHAM6.1, and the negative shortwave cloud term also contributes to lesser warming in ECHAM6.1.

In the Arctic, the surface temperature warming is large in both models in the Northern hemispheric autumn and winter and characterized by a near lack of negative energetic terms and strong LW$_{clr}$ terms in both models. The models produce mixed results for the Arctic spring, but both models show a steep summer minimum in the overall surface temperature response. The Arctic summer response is characterized by the positive surface albedo (ALBEDO) and energy transport effects (CONV) opposed by a strongly negative surface energy term (SURF) corresponding to oceanic heat uptake. In Arctic summer, the

shortwave cloud effects SW$_{cld}$ are more negative in ECHAM6.1 than in NorESM1, with very modest effects for the rest of the year. During other seasons, the surface energy (SURF) term becomes positive as the ocean releases the energy stored. Thus, in the Arctic, SURF amplifies the seasonality of the temperature response.

## 3 Conclusions and discussion

In this work, we have represented the modern day anthropogenic aerosols identically in two climate models with independent

development histories, and studied the equilibrium climate responses to the removal of East and South Asian anthropogenic aerosols. This forcing gives rise to a positive surface temperature response, the global mean of which is somewhat larger in NorESM1 (0.30±0.03 K) than in ECHAM6.1 (0.22±0.03 K). Both models robustly show that the warming response spreads across both hemispheres and is particularly strong in the Arctic.

The temperature decomposition method by Räisänen (2017) provides a valuable tool for analyzing how the surface temperature response to a regional forcing spreads to remote regions. Over the polluted regions in South and East Asia, the removal of



anthropogenic aerosols leads to a strong surface warming contribution from additional solar radiation reaching the surface under clear-sky conditions. The local temperature effects due to changes in clouds are however more complex. While the removal of modeled aerosols in the applied MACv2-SP aerosol scheme (Stevens et al., 2017) only affects the cloud shortwave
properties via the first indirect aerosol effect, the cloud responses manifest themselves both in shortwave and longwave channels, with changes in cloud amount having opposite shortwave and longwave effects on surface temperatures.

The driver of the wide geographical spreading of the temperature response appears to be the strong tendency of atmospheric heat transport to regulate surface warming over the region of diminished aerosol forcing while simultaneously enhancing the
warming in remote locations. Further, changes in the clear-sky longwave responses spread the surface temperature warming over both hemispheres. In both models the seasonality of the latitudinal surface temperature responses is modest in northern low and mid-latitudes, but strong over the Arctic.

The mechanisms driving the strongly seasonal Arctic response resembles those for the CMIP5 ensemble for $CO_2$ doubling
(Räisänen, 2017) and historical transient simulations (Pithan and Mauritsen, 2014). They involve the ice-albedo feedback, where additional sea ice melt during spring and summer leads to increased absorption of solar radiation by the larger open water area in the Arctic Ocean during summer and autumn. During the summer the Arctic Ocean is thermalized close to the freezing temperature and the trapped solar radiation is stored as heat within the ocean (e.g. Holland and Bitz, 2003; Screen and Simmonds, 2010). This heat is then released during autumn and winter, elevating the atmospheric sub-zero temperatures.
However, the longwave clear-sky response contributes to the seasonality and the overall Arctic warming, supporting the strong role of temperature feedbacks in the Arctic warming (Pithan and Mauritsen, 2014) also in case of South and East Asian aerosol removal. Further, it is notable that in this study a strong Arctic surface temperature response takes place despite the lack of modeled changes in oceanic heat transport, which have been previously shown to dominate the increase in heat transport towards the Arctic due to reductions in European anthropogenic aerosol emissions (Acosta-Navarro et al., 2016).

The temperature decomposition method also allows an analysis of the similarities and differences between the response in ECHAM6.1 and NorESM1. It was found that the larger global annual mean warming in NorESM1 than in ECHAM6.1 (0.30±0.03 K vs. 0.22±0.03 K) is primarily associated with the shortwave cloud response (0.07±0.02 K for NorESM1 and 0.03±0.03 K for ECHAM6.1) and the longwave cloud response (0.05±0.01 K for NorESM1 and 0.01±0.01 K for ECHAM6.1).
Furthermore, there are significant differences in the geographic patterns of cloud cover responses, which lead to equally significant local/regional differences in the combined shortwave and longwave cloud surface temperature responses. Overall, these differences notwithstanding, it is however encouraging that the geographical distribution of remote surface temperature response is robust in the two independent climate models, when run with identical aerosol descriptions. Not just the distribution of total surface temperature response is similar in the models, but also the distributions of different energy flux
drivers which together constitute the obtained temperature responses, are mostly similar.

The effective radiative forcing (ERF) due to *adding* MACv2-SP aerosols was shown to be -0.50 Wm$^{-2}$ for ECHAM6.3 and -0.65 Wm$^{-2}$ for NorESM1 by Fiedler et al. (2019). As such, the total ERF for all anthropogenic aerosols computed using the MACv2-SP scheme is in the low-end range of typical ERFs (between -0.29 and -1.44 Wm$^{-2}$) obtained for CMIP5 models with

model-intrinsic aerosol schemes (Shindell et al., 2015), and closely matches the recent estimate of -0.55 Wm$^{-2}$ for the 1750-2015 change in global aerosol ERF by Lund et al. (2019).  The global annual temperature response for adding MACv2-SP aerosols was shown to be -0.48K for ECHAM6.1 and -0.50K for NorESM1, being in the low end of equilibrium temperature responses (-0.5 to -1.1K) for adding model-intrinsic anthropogenic aerosols in four contemporary climate models (Samset et al., 2018). Therefore, the annual average temperature response of 0.26±0.04 K obtained here can be considered to be a

conservative estimate for the removal of South and East Asian anthropogenic aerosols.

To contextualize the effects of strong Asian aerosol pollution mitigation scenarios on the changes in global surface temperatures, we note that global temperatures have increased by an average of 0.18 °C per decade during 1980-2019 (NOAA global climate report 2019). Lund et al. (2019) showed that under the Socioeconomic Shared Pathway 1-1.9, the strong air

pollution mitigation scenarios tied with $CO_2$ mitigation policies lead to a 55% drop in combined aerosol emissions from South and East Asian regions already by 2030. Our models predict an annually averaged global warming of 0.26±0.04 °C  if the South and East Asian anthropogenic aerosols are removed totally. Assuming a linear relationship between aerosol emission reductions and temperature effects and a relatively fast transient climate response for the aerosols, the Asian emissions reductions can add another 7.9 (6.7-9.2) years' worth of current day global warming on top of greenhouse gas -related warming

during the next few decades, thus significantly pushing back the near-decadal effects of strong $CO_2$ mitigation policies under Socioeconomic Shared Pathway 1-1.9.

**Data availability**

Data and scripts are available at https://fmi.b2share.csc.fi/records/aa3085799bff4be798342e4c0d66caa3.

**Author contributions**

The manuscript was written by JM with contributions from all authors. KN performed ECHAM6.1 simulations with help from JM, PR and DO'D. PR performed all NorESM1 simulations. The data analysis was carried out by JM and KN and guided by JR. HK came up with the initial research idea and AIP assisted with project coordination.





**Competing interests**

The authors declare no competing interests.

**Acknowledgements**

This project has been funded by the European Research Council (ERC) under the European Union's Horizon 2020 research and innovation program under grant agreement no. 646857, and by the Academy of Finland (projects 287440 and 308365). The authors would also like to thank Stephanie Fiedler for providing the MACv2-SP code for ECHAM6.1.

**Financial support**

This research has been supported by the Academy of Finland (grant no. 287440) and the European Research Council (grant no. 646857).

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



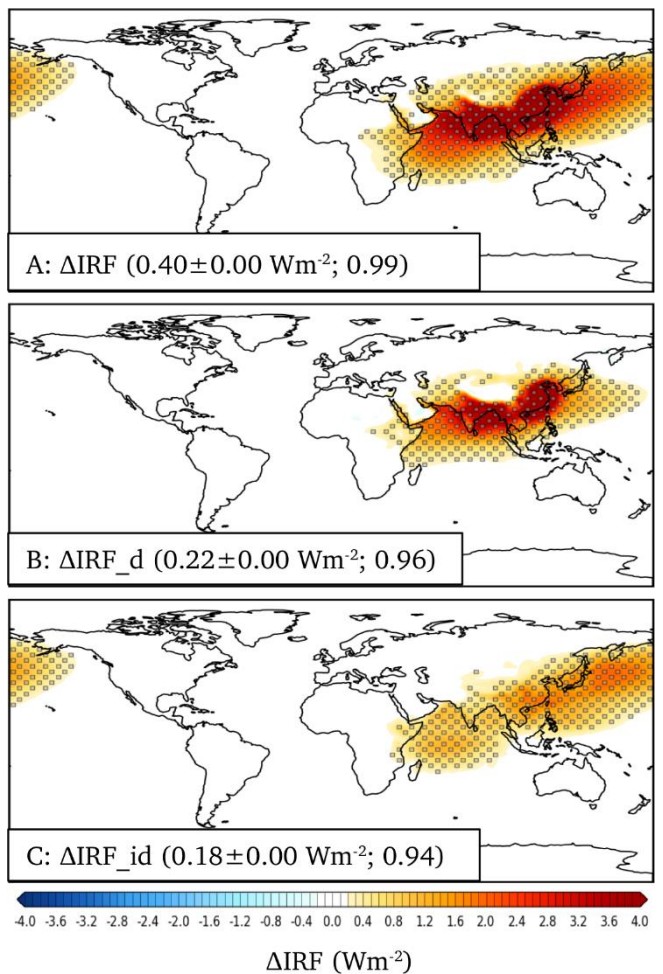

**Figure 1: The change in the mean instantaneous radiative forcing between runs without and with SE Asian aerosols using the MACv2-SP aerosol scheme, averaged over the two climate models (individual models shown in Fig. S1). Parenthesis show the average global mean value and the model-to-model correlation coefficient, respectively. A: Change in the total aerosol radiative forcing, B: change in the direct aerosol radiative forcing, and C: change in the indirect aerosol radiative forcing. Stippling shows regions where the results are statistically significant at the p<0.05 level for both models, and models also agree on the sign.**



**Figure 2: The geographical distributions of annually averaged surface air (2m) temperature responses due to the removal of South and East Asian aerosols (mean over ECHAM6.1 and NorESM1 climate models). Brackets show global average responses in Kelvins and the model-to-model correlation coefficient, respectively. Panel A: the total surface temperature response. Panels B-H: Contributions to the total surface temperature response from shortwave clear-sky response (B), horizontal atmospheric heat transport (C), shortwave cloud response (D), longwave cloud response (E), surface albedo change (F), longwave clear-sky response (G), and surface energy flux change (H). Panels B-H sum up to the response seen in Panel A. Panel I shows the combined shortwave and longwave cloud response. Stippling shows regions where the results are statistically significant at the p<0.05 level for both models, and models also agree on the sign.**






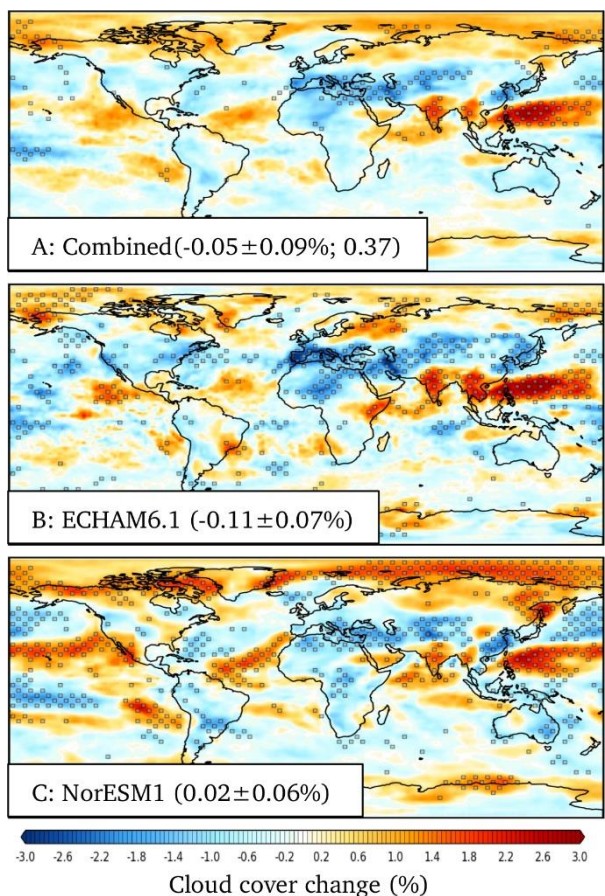

Figure 3: The geographical distributions of annually averaged changes in total cloud cover due to the removal of South and East Asian aerosols. A: Cloud cover change averaged over both ECHAM6.1 and NorESM1 climate models. Brackets show global average responses in percentages and the model-to-model correlation coefficient, respectively. Stippling shows regions where the results are statistically significant at the $p<0.05$ level for both models, and models also agree on the sign. Panel B and C: Cloud cover change in ECHAM6.1 and NorESM1, respectively. Brackets give global average responses in percentages. Stippling shows regions where the results are statistically significant at the $p<0.05$ level.



**Figure 4: Seasonal cycles of surface temperature responses averaged over the Northern hemispheric latitude bands for the ECHAM6.1 (A) and NorESM1 (B) models. Color bars show the different contributions to seasonal mean temperature responses shown with the star symbols. Different seasons are indicated with different hatchings over the color bars. The solid lines indicate the annual average Asian aerosol radiative forcing RF for each model. The modest seasonality in radiative forcing is not shown due to sake of clarity.**





**Appendix**

ECHAM6.1         NorESM1

ΔIRF (0.38±0.00 Wm⁻²)    ΔIRF (0.41±0.00 Wm⁻²)

ΔIRF_d (0.23±0.00 Wm⁻²)    ΔIRF_d (0.20±0.00 Wm⁻²)

ΔIRF_id (0.15±0.00 Wm⁻²)    ΔIRF_id (0.21±0.00 Wm⁻²)

-4.0 -3.6 -3.2 -2.8 -2.4 -2.0 -1.6 -1.2 -0.8 -0.4 0.0 0.4 0.8 1.2 1.6 2.0 2.4 2.8 3.2 3.6 4.0

ΔIRF (Wm⁻²)

**Figure A1: The change in the mean instantaneous radiative forcings for runs without and with SE Asian aerosols shown for the
individual models (left panels, ECHAM6; right panels, NorESM).**




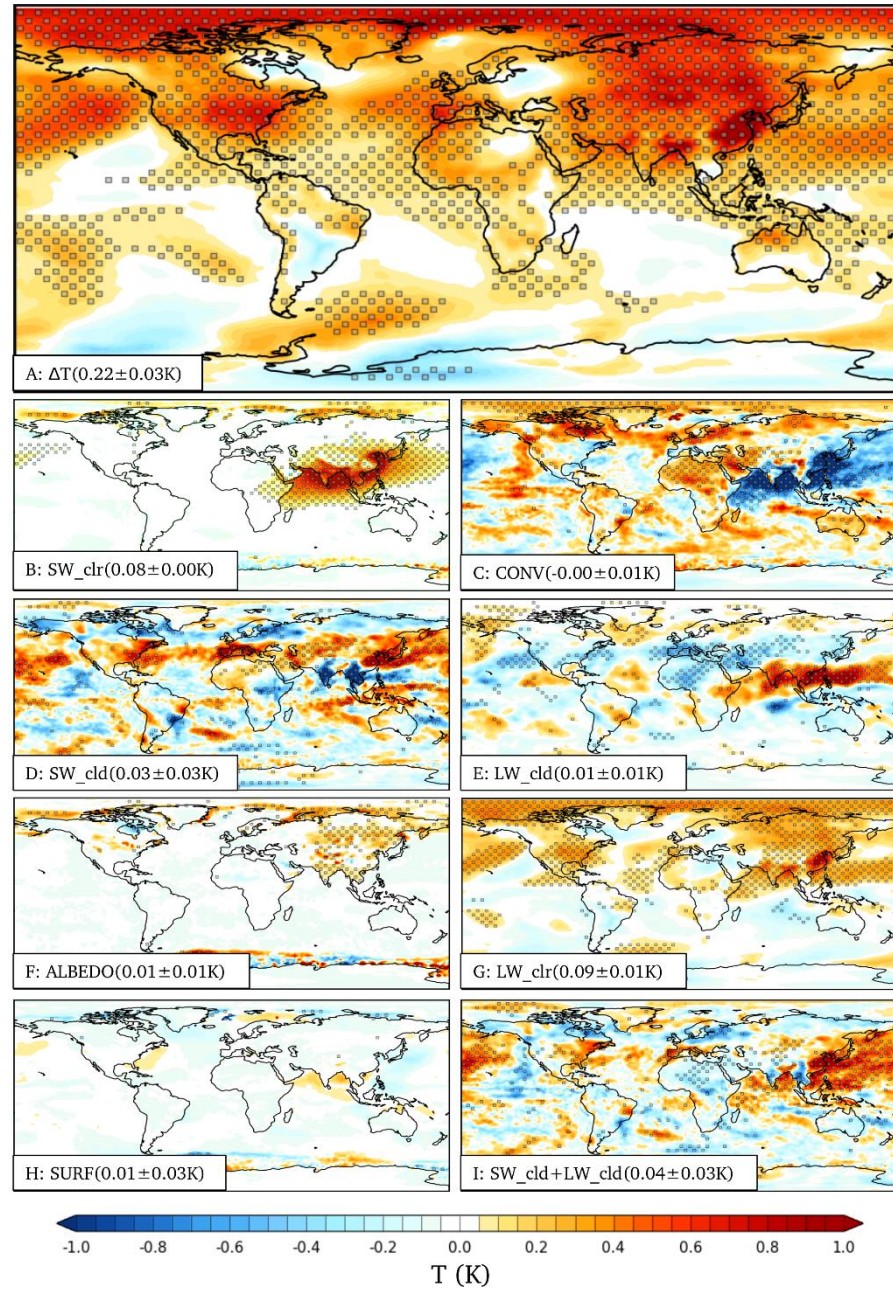

**Figure A2: The geographical distributions of annually averaged surface air (2m) temperature responses due to the removal of South and East Asian aerosols for ECHAM6. Brackets show global averages. Panel A: the total surface responses. Panels B-H: Contributions to the total surface temperature response from shortwave clear-sky response (B), horizontal atmospheric heat transport (C), shortwave cloud response (D), longwave cloud response (E), surface albedo change (F), longwave clear-sky response (G), and surface energy flux change (H). Panels B-H sum up to the response seen in Panel A. Panel I shows the combined shortwave and longwave cloud response.**






**Figure A3: As Fig A2, but for the NorESM model**




| | $\Delta T$ | $LW_{clr}$ | $SW_{clr}$ | $LW_{cld}$ | $SW_{cld}$ | ALBEDO | CONV | SURF | $LW_{cld} + SW_{cld}$ |
|---|---|---|---|---|---|---|---|---|---|
| ECHAM6.1 | 0.22±0.03 | 0.09±0.01 | 0.08±0.00 | 0.01±0.01 | 0.03±0.03 | 0.01±0.01 | -0.00±0.01 | 0.01±0.03 | 0.04±0.03 |
| | (0.12876) | (0.05751) | (0.01674) | (0.03961) | (0.12634) | (0.03652) | (0.03005) | (0.09962) | (0.01169) |
| NorESM1 | 0.30±0.03 | 0.08±0.02 | 0.09±0.00 | 0.05±0.01 | 0.07±0.02 | 0.03±0.01 | -0.01±0.01 | 0.00±0.03 | 0.12±0.02 |
| | (0.11558) | (0.08762) | (0.01489) | (0.04337) | (0.09526) | (0.02346) | (0.05439) | (0.09717) | (0.07647) |
| Model mean | 0.26±0.04 | 0.08±0.03 | 0.08±0.01 | 0.03±0.02 | 0.05±0.04 | 0.02±0.01 | -0.01±0.02 | 0.00±0.04 | 0.08±0.04 |
| | (0.12235) | (0.07411) | (0.01584) | (0.04153) | (0.11188) | (0.03069) | (0.04394) | (0.09840) | (0.09879) |
| Correlation | 0.651 | 0.519 | 0.918 | 0.461 | 0.279 | 0.226 | 0.642 | -0.031 | 0.372 |

**Table A1: Upper rows for each model and model mean: yearly global mean values in Kelvins, with errors on the means given with a 95% confidence interval. Error on the model mean is given as a pooled sample standard error on the mean. Values in brackets show the standard deviations in yearly mean values (pooled standard deviations for model mean). The bottom row: spatial correlation between NorESM1 and ECHAM6.1.**
