# Peer review of "How Asian aerosols impact regional surface temperatures across the globe"

_Atmospheric Chemistry and Physics, 2020_

## Referee Comment (RC1) · Anonymous Referee #1 · 9 Jan 2021

General comments:

The authors analyse slab-ocean simulations with two different climate models (NorESM1 and ECHAM6.1) to understand the global surface temperature responses to a 100% reduction in Asian anthropogenic aerosol emissions. The manuscript adds to a growing body of literature investigating climate responses to regional aerosol emissions, increasingly recognised as important because of the heterogeneous nature of the forcing, and the additional warming which is anticipated as Asian aerosol emissions in particular are projected to decline steeply over coming decades.

Although other multi-model studies have previously investigated removing Asian aerosol emissions, the present study goes further by describing a method to decompose the global pattern and seasonal cycle of surface temperature changes into contributions from different energy fluxes. The novel application of this methodology to understand the global response to a regionally-localised radiative forcing in a complex climate model, provides considerable insight into the contributions of different atmospheric processes in redistributing heat to create a hemisphere-wide surface temperature response.

The manuscript is timely, well written and presented, and certainly of interest to the audience of this journal. I really only have a small number of very minor comments and technical corrections listed below that I would like to see the authors address, and subject to these I would recommend it be accepted for publication in Atmospheric Chemistry and Physics.

Specific comments:

Introduction: The authors could consider to also reference the study by Liu et al. (J. Clim 2018, https://doi.org/10.1175/JCLI-D-17-0439.1) which looked at patterns of climate response to a regional Asian aerosol perturbation in multiple models, including by performing a breakdown of the response into different energy budget terms (though far less comprehensively than in this study) - predominantly with regard to understanding the precipitation change although the temperature responses are also discussed

L91: By "background aerosol" I assume this refers to natural aerosol sources (e.g. dust, sea salt)? It might be useful just to explicitly say this here (e.g. "The background *natural* aerosols..." or something similar), so it's clear that only natural aerosols are represented differently between the two models.

L133-134: consider mentioning that that SH and LH are the *net downwards* sensible and latent heat fluxes (at least this is what they seem to be, from the sign of the terms in the equation), since this is opposite to the conventional sign of these terms which are more commonly defined as net upwards

L167-168: "We mark ... collectively as CONV" - consider adding something like "as

together they represent the convergence of energy" or something similar, so that it is clear where the abbreviation CONV comes from

L390-391: "changes in the clear-sky longwave responses spread the surface temperature warming over both hemispheres" - maybe I haven't understood the plots properly, but based on Fig 2 this statement doesn't seem right. Looking at Fig 2G, it appears to me that the LW_clr term is highly restricted to the northern hemisphere, and is mostly zero or slightly negative in the southern hemisphere. In fact it appears to be one of the few terms which *doesn't* contribute much to the southern hemisphere response. But maybe I've misunderstood the figure here, so please correct me if so! On a related note, if LW_clr is indeed the main term responsible for spreading the response to both hemispheres, there would seem to be a chicken-and-egg question of causality: Earlier in the manuscript I think the authors attribute the LW_clr response to water vapour and lapse-rate feedbacks, but presumably this requires there to first be some initial warming due to another process. I realise these are equilibrium responses so it is hard to diagnose, but again it seems counter-intuitive that LW_clr could be the main redistributor into the southern hemisphere unless it's the feedback to another term which is already moving heat into the southern hemisphere. Do the authors have any hypothesis what that initial process might be? (Again I understand this might be hard to determine from these simulations, mainly I'm curious just to satisfy myself that there's a plausible hypothesis)

L400: From Fig 4, it looks to me like the LW_clr term actually contributes more than the albedo term to both the seasonality and the total Arctic amplification (which incidentally is the same as Pithan and Mauritsen find). However the current wording makes it sound like the LW_clr term is secondary to the ice-albedo feedback. Maybe say something like "However, the longwave clear-sky response contributes *even more* to the seasonality and the overall Arctic warming" or something similar? Again, unless I have misinterpreted Fig 4 in which case please correct me!

Other technical corrections:

L38 and L54: Reference misspelt, should be 'Westervelt'

L43: Reference should be Lamarque et al., not just Lamarque

L98: There appears to be a missing word - I assume it should say "aerosol radiative forcings" or "aerosol radiaitve effects" or something similar

L102: Typo - "NoreSM1" should say "NorESM1"

L334: KK -> K

L597 & L643: Kelvins -> kelvin or kelvins (lower case k, pluralisation optional)

---

## Referee Comment (RC2) · Anonymous Referee #2 · 4 Feb 2021

Review on "How Asian aerosols impact regional surface temperatures across the globe" by Merikanto et al.

General:

This manuscript investigates the global temperature response to the removal of anthropogenic aerosol in S&E Asia, by running two global climate models coupled with the slab ocean. The novelty of this work is a decomposition of the global temperature responses into contributions from multiple energy components, which help to identify the sources of temperature responses to different physical processes. I think the manuscript is well-organized and neatly-written. The figures and tables are fully supporting the conclusions. Therefore, I would suggest accepting the manuscript after minor revisions on a few points for clarification.

Specific comments:

1) Line 64, Line 243, Line 402 and perhaps somewhere else

   "removal of South and East Asian aerosols". I think this study removes ANTHROPOGENIC aerosols in S&E Asia, not ALL aerosols in S&E Asia. Please be more precise in the context.

2) Line 98:

   What is "indirect instantaneous aerosol radiative"?

3) Line 107-109:

What does the "natural background aerosol" exactly mean? Sulfate from DMS over ocean? Carbonaceous aerosols from natural sources such as wildfire? I think the species and brief info about sources of "natural background aerosol" or "background aerosol" should be specified at least.

4) Section 2.1:

As the manuscript focuses on surface temperature response to the radiative forcing of anthropogenic aerosols in S&E Asia. I would be curious about what the climate sensitivities of the two models are. Climate sensitivity is essentially related to water vapor feedback, cloud feedback and ice-albedo feedback etc. I think knowing the climate sensitivities of the two models would help the audiences better understand how sensitive the surface temperature is responded to different physical processes (especially the cloud-related process).

5) Line 183:

"such as to changes in atmospheric and surface temperature AND/OR? water vapor"

6) Line 237-241:

I do not get the point quite well here. What do you mean the "cancellation of differences in $\Delta$IRF"? Is it referring to Figure A1 that $\Delta$IRF$_d$ in ECHAM6.1 is stronger than that in NorESM1 but $\Delta$

$IRF_{id}$ in ECHAM6.1 is weaker than that in NorESM1. $\Delta IRF$ is obtained by summing up $\Delta IRF_d$ and $\Delta IRF_{id}$, thus $\Delta IRF$ in the two models have more similar distributions and has higher model-to-model correlation coefficient than $\Delta IRF_d$ and $\Delta IRF_{id}$ respectively. Is it correct?

7) Line 323-324:

Why does emphasize the similar cc for $SW_{cld}+LW_{cld}$ and for total cloud cover here? I see from Figs. 3A and 2E (Figs. 3B and A2E; Figs. 3C and A3E) that the distribution of total cloud cover is more similar to distribution of $LW_{cld}$, not the distribution of $SW_{cld}+LW_{cld}$. Is it correct?

---

## Author Comment (AC2) · 23 Feb 2021

Response to Reviewer 2:

We thank the reviewer for the valuable comments on the manuscript. We have revised the manuscript according to the suggestions. Below, we itemize the original review comments and the changes made to the manuscript.

Comment 1: Line 64, Line 243, Line 402 and perhaps somewhere else "removal of South and East Asian aerosols". I think this study removes ANTHROPOGENIC aerosols in S&E Asia, not ALL aerosols in S&E Asia. Please be more precise in the context.

Reply 1: Throughout the text we now refer to "S&E Asian anthropogenic aerosols".

Comment 2: Line 98: What is "indirect instantaneous aerosol radiative"?

Reply 2: There was a word missing from the sentence. It now reads "indirect instantaneous aerosol radiative forcings".

Comment 3: Line 107-109: What does the "natural background aerosol" exactly mean? Sulfate from DMS over ocean? Carbonaceous aerosols from natural sources such as wildfire? I think the species and brief info about sources of "natural background aerosol" or "background aerosol" should be specified at least.

Reply 3: We agree that the description was vague. While the background aerosols in both models are mostly of natural origin, they actually aim to represent the pre-industrial aerosols. This issue is more thoroughly discussed in Fiedler et al. (2019), to which we now also refer to: L101 in revised MS: "The background pre-industrial aerosols (mainly consisting of natural organics and sulfate, sea salt and dust) for ECHAM6.1 are prescribed using the climatology of Kinne et al. (2013), while for NorESM1, they are simulated by the model's bottom-up aerosol microphysics scheme (Kirkevåg et al. 2013) (see also Fig. 2 and Appendix A in Fiedler et al. (2019) describing the pre-industrial aerosols for both of the applied models, and the related discussion within)."

Comment 4: Section 2.1: As the manuscript focuses on surface temperature response to the radiative forcing of anthropogenic aerosols in S&E Asia. I would be curious about what the climate sensitivities of the two models are. Climate sensitivity is essentially related to water vapor feedback, cloud feedback and ice-albedo feedback etc. I think knowing the climate sensitivities of the two models would help the audiences better understand how sensitive the surface temperature is responded to different physical processes (especially the cloud-related process).

Reply 4: We thank the reviewer for this suggestion, as it helps to further contextualize the results. We added the following text and references to Section 2.1: L123 in revised MS: "The reported equilibrium climate sensitivity is 3.5K for NorESM1 (Räisänen et al.,

2017) and also 3.5K for ECHAM6.1 (Mauritsen and Roeckner, 2020)."

Comment 5: Line 183: "such as to changes in atmospheric and surface temperature AND/OR? water vapor"

Reply 5: We thank the reviewer for spotting this typo. The line now reads: L197 in revised MS: "such as to changes in atmospheric temperature, surface temperature, or water vapor under clear-sky and all-sky conditions."

Comment 6: Line 237-241: I do not get the point quite well here. What do you mean the "cancellation of differences in $\Delta$IRF"? Is it referring to Figure A1 that $\Delta$IRFd in ECHAM6.1 is stronger than that in NorESM1 but $\Delta$IRFid in ECHAM6.1 is weaker than that in NorESM1. $\Delta$IRF is obtained by summing up $\Delta$IRFd and $\Delta$IRFid, thus $\Delta$IRF in the two models have more similar distributions and has higher model-to-model correlation coefficient than $\Delta$IRFd and $\Delta$IRFid respectively. Is it correct?

Reply6: This is correct. We agree that the explanation was unclear. We now clarify this "compensation of differences" by adding a sentence: L255 in revised MS: "While the aerosols enhance the cloud albedo, clouds also diminish the direct reflection of sunlight by aerosols with compensating effects on the total radiative response."

Comment 7: Line 323-324: Why does emphasize the similar cc for SWcld+LWcld and for total cloud cover here? I see from Figs. 3A and 2E (Figs. 3B and A2E; Figs. 3C and A3E) that the distribution of total cloud cover is more similar to distribution of LWcld, not the distribution of SWcld+LWcld. Is it correct?

Reply 7: This is correct. However, in the section at question (lines 323-324) we focus on the model-to-model differences, and mention that the total cloud response SWcld+LWcld between the two models correlates as "poorly" (cc=0.37) as the total cloud cover change between the two the models (0.37). The cloud cover change does have a rather high correlation with LWcld (cc=0.77) and a rather high anti-correlation (with signs being opposite) with SWcld (-0.74), as discussed in the section before (line

279). To emphasize that here we discuss the model-to-model differences, we modified the sentence in question to: L342 in revised MS: "The total surface temperature response due to clouds in the two models, SWcld+ LWcld (cc=0.37) has a similarly low correlation as the change in total cloud cover (cc=0.37) between the two models."

………………………………………………………………………………………

References:

Mauritsen, T., and Roeckner, E.: Tuning the MPI-ESM1.2 global climate model to improve the match with instrumental record warming by lowering its climate sensitivity, Journal of Advances in Modeling Earth Systems, 12, e2019MS002037, https://doi.org/10.1029/2019MS002037, 2020.

Räisänen, P., Makkonen, R., Kirkevåg, A., and Debernard, J. B.: Effects of snow grain shape on climate simulations: sensitivity tests with the Norwegian Earth System Model, The Cryosphere, 11, 2919–2942, https://doi.org/10.5194/tc-11-2919-2017, 2017.

---

## Author Response (AR1)

Replies to Reviewer 1:

We would like to thank the reviewer for the constructive comments and suggestions. Below we reply to every comment made by the reviewer. The original comments are numerated and repeated in blue font below, followed by our answers.

Specific comments:

*C1: Introduction: The authors could consider to also reference the study by Liu et al. (J.Clim 2018, https://doi.org/10.1175/JCLI-D-17-0439.1) which looked at patterns of climate response to a regional Asian aerosol perturbation in multiple models, including by performing a breakdown of the response into different energy budget terms (though far less comprehensively than in this study) – predominantly with regard to understanding the precipitation change although the temperature responses are also discussed.*

We have added the reference to Liu et al. (2018) and discuss the relevant findings in that paper in the Introduction and in Results sections. Specifically, we added:

L65 in revised MS: "Liu et al. (2018) showed that the temperature effects of idealized Asian aerosol perturbations spread across the Northern hemisphere in a multi-model PDRMIP study, and that increases in Asian sulfate aerosols strongly suppressed Asian monsoon precipitation by enhancing horizontal atmospheric heat transport to the region and raising surface pressure."

L270 in revised MS: "A similar global climate sensitivity of $0.58\pm0.23$ K/(Wm$^{-2}$) for a 10-fold increase in Asian anthropogenic sulfate aerosols was found in models that participated to the multi-model intercomparison project PDRMIP (Liu et al., 2018)."

*C2: L91: By "background aerosol" I assume this refers to natural aerosol sources (e.g.dust, sea salt)? It might be useful just to explicitly say this here (e.g. "The background \*natural\* aerosols…" or something similar), so it's clear that only natural aerosols are represented differently between the two models.*

We discuss the background aerosols now in more detail:
L101 in revised MS: "The background pre-industrial aerosols (mainly consisting of natural organics and sulfate, sea salt and dust) for ECHAM6.1 are prescribed using the climatology of Kinne et al. (2013), while for NorESM1, they are simulated by the model's bottom-up aerosol microphysics scheme (Kirkevåg et al. 2013) (see also Fig. 2 and Appendix A in Fiedler et al. (2019) describing the pre-industrial aerosols for both of the applied models, and the related discussion)."

*C3: L133-134: consider mentioning that that SH and LH are the \*net downwards\* sensible and latent heat fluxes (at least this is what they seem to be, from the sign of the terms in the equation), since this is opposite to the conventional sign of these terms which are more commonly defined as net upwards.*

We thank the reviewer for pointing out the need for this clarification. Indeed, the net sensible and latent heat fluxes were taken as "net downwards" fluxes. We have now replaced "$SH$" and "$LH$" by "$-SH^{\uparrow}$" and "$-LH^{\uparrow}$" to follow the conventional sign convention.

*C4: L167-168: "We mark … collectively as CONV" – consider adding something like "as together they represent the convergence of energy" or something similar, so that it is clear where the abbreviation CONV comes from.*

Added.

*C5: L390-391: "changes in the clear-sky longwave responses spread the surface temperature warming over both hemispheres" - maybe I haven't understood the plots properly, but based on Fig 2 this statement doesn't seem right. Looking at Fig 2G, it appears to me that the LW_clr term is highly restricted to the northern hemisphere, and is mostly zero or slightly negative in the southern hemisphere. In fact it appears to be one of the few terms which \*doesn't\* contribute much to the southern hemisphere response. But maybe I've misunderstood the figure here, so please correct me if so! On a related note, if LW_clr is indeed the main term responsible for spreading the response to both hemispheres, there would seem to be a chicken-and-egg question of causality: Earlier in the manuscript I think the authors attribute the LW_clr response to water vapour and lapse-rate feedbacks, but presumably this requires there to first be some initial warming due to another process. I realise these are equilibrium responses so it is hard to diagnose, but again it seems counter-intuitive that LW_clr could be the main redistributor into the southern hemisphere unless it's the feedback to another term which is already moving heat into the southern hemisphere. Do the authors have any hypothesis what that initial process might be? (Again I understand this might be hard to determine from these simulations, mainly I'm curious just to satisfy myself that there's a plausible hypothesis).*

It is indeed true that LW_clr is mainly restricted to Northern hemisphere, and this result also applies to both applied models separately. Hence, the original claim that "clear-sky longwave responses spread the surface temperature warming over both hemispheres" was inaccurately formulated. On the related note, we see the changes in heat transport as the primary driver of the remote feedbacks, while the full disentangling of feedbacks from responses is difficult. But we could envision that large scale circulation changes (such as a shift in ITCZ) might also lead to changes in LW_clr.

We changed the sentence to:
L409 in revised MS: "The driver of the wide geographical spreading of the temperature response appears to be the strong tendency of atmospheric heat transport to regulate surface warming over the region of diminished aerosol forcing while simultaneously enhancing the warming in remote locations. Also, changes in the clear-sky longwave responses associated at least in part with increased water vapor further amplify the surface temperature warming over the Northern hemisphere."

*C6: L400: From Fig 4, it looks to me like the LW_clr term actually contributes more than the albedo term to both the seasonality and the total Arctic amplification (which incidentally is the same as Pithan and Mauritsen find). However the current wording makes it sound like the LW_clr term is secondary to the ice-albedo feedback. Maybe say something like "However, the longwave clear-sky response contributes \*even more\* to the seasonality and the overall Arctic warming" or something similar? Again, unless I have misinterpreted Fig 4 in which case please correct me!*

Again we agree, and now we emphasize the role of LW_clr as the main contributor of the Arctic temperature response. We modified the sentence to:
L422 in revised MS: "However, it is the longwave clear-sky response that contributes most to the seasonality and the overall Arctic warming, supporting the strong role of temperature feedbacks in the Arctic warming (Pithan and Mauritsen, 2014) also in case of South and East Asian anthropogenic aerosol removal."

*Other technical corrections:*

*C7: L38 and L54: Reference misspelt, should be 'Westervelt'.*

Corrected.

*C8: L43: Reference should be Lamarque et al., not just Lamarque.*

Corrected.

C9: L98: There appears to be a missing word - I assume it should say "aerosol radiative forcings" or "aerosol radiative effects" or something similar

Corrected to "aerosol radiative forcings".

C10: L102: Typo - "NoreSM1" should say "NorESM1"

Corrected.

C11: L334: KK -> K

Corrected.

C12: L597 & L643: Kelvins -> kelvin or kelvins (lower case k, pluralisation optional).

Corrected to kelvins.

………………………………………………………………………………………..

Response to Reviewer 2:

We thank the reviewer for the valuable comments on the manuscript, and we have revised the manuscript according to the suggestions. Below, we itemize the original comments (in blue font) and the changes made to the manuscript.

*1) Line 64, Line 243, Line 402 and perhaps somewhere else "removal of South and East Asian aerosols". I think this study removes ANTHROPOGENIC aerosols in S&E Asia, not ALL aerosols in S&E Asia. Please be more precise in the context.*

Throughout the text we now refer to "S&E Asian anthropogenic aerosols".

*2) Line 98: What is "indirect instantaneous aerosol radiative"?*

There was a word missing from the sentence. It now reads "indirect instantaneous aerosol radiative forcings".

*3) Line 107-109: What does the "natural background aerosol" exactly mean? Sulfate from DMS over ocean? Carbonaceous aerosols from natural sources such as wildfire? I think the species and brief info about sources of "natural background aerosol" or "background aerosol" should be specified at least.*

We agree that the description was vague. While the background aerosols in both models are mostly of natural origin, they actually aim to represent the pre-industrial aerosols. This issue is more thoroughly discussed in Fiedler et al. (2019), to which we now also refer to:

L101 in revised MS: "The background pre-industrial aerosols (mainly consisting of natural organics and sulfate, sea salt and dust) for ECHAM6.1 are prescribed using the climatology of Kinne et al. (2013), while for NorESM1, they are simulated by the model's bottom-up aerosol microphysics scheme (Kirkevåg et al. 2013) (see also Fig. 2 and Appendix A in Fiedler et al. (2019) describing the pre-industrial aerosols for both of the applied models, and the related discussion)."

*4) Section 2.1: As the manuscript focuses on surface temperature response to the radiative forcing of anthropogenic aerosols in S&E Asia. I would be curious about what the climate sensitivities of the two models are. Climate sensitivity is essentially related to water vapor feedback, cloud feedback and ice-albedo feedback etc. I think knowing the climate sensitivities of the two models would help the audiences better understand how sensitive the surface temperature is responded to different physical processes (especially the cloud-related process).*

We thank the reviewer for this suggestion, as it helps to further contextualize the results. We added the following text to Section 2.1:

L123 in revised MS: "The reported equilibrium climate sensitivities are 3.5K for NorESM1 (Räisänen et al., 2017) and also 3.5K for ECHAM6.1 (Mauritsen and Roeckner, 2020)."

*5) Line 183: "such as to changes in atmospheric and surface temperature AND/OR? water vapor"*

We thank the reviewer for spotting this typo. The line now reads:

L197 in revised MS: "such as to changes in atmospheric temperature, surface temperature, or water vapor under clear-sky and all-sky conditions."

*6) Line 237-241: I do not get the point quite well here. What do you mean the "cancellation of differences in ΔIRF"? Is it referring to Figure A1 that $\Delta IRF_d$ in ECHAM6.1 is stronger than that in NorESM1 but $\Delta IRF_{id}$ in ECHAM6.1 is weaker than that in NorESM1. ΔIRF is obtained by summing up $\Delta IRF_d$ and $\Delta IRF_{id}$, thus ΔIRF in the two models have more similar distributions and has higher model-to-model correlation coefficient than $\Delta IRF_d$ and $\Delta IRF_{id}$ respectively. Is it correct?*

This is correct. We agree that the explanation was unclear. We now clarify this "compensation of differences" by adding a sentence:

L255 in revised MS: "While the aerosols enhance the cloud albedo, clouds also diminish the direct reflection of sunlight by aerosols with compensating effects on the total radiative response."

*7) Line 323-324: Why does emphasize the similar cc for $SW_{cld}+LW_{cld}$ and for total cloud cover here? I see from Figs. 3A and 2E (Figs. 3B and A2E; Figs. 3C and A3E) that the distribution of total cloud cover is more similar to distribution of $LW_{cld}$, not the distribution of $SW_{cld}+LW_{cld}$. Is it correct?*

This is correct. However, in the section at question (lines 323-324) we focus on the model-to-model differences, and mention that the total cloud response $SW_{cld}+LW_{cld}$ between the two models correlates as "poorly" (cc=0.37) as the total cloud cover change between the two the models (0.37). The cloud cover change does have a rather high correlation with $LW_{cld}$ (cc=0.77) and a rather high anti-correlation (with signs being opposite) with $SW_{cld}$ (-0.74), as discussed in the section before (line 279). To emphasize that here we discuss the model-to-model differences, we modified the sentence in question to:

L342 in revised MS: "The total surface temperature response due to clouds in the two models, $SW_{cld}+ LW_{cld}$ (cc=0.37) has a similarly low correlation as the change in total cloud cover (cc=0.37) between the two models."

………………………………………………………………………………………………………………………

References:

Mauritsen, T., and Roeckner, E.: Tuning the MPI-ESM1.2 global climate model to improve the match with instrumental record warming by lowering its climate sensitivity, Journal of Advances in Modeling Earth Systems, 12, e2019MS002037, https://doi.org/10.1029/2019MS002037, 2020.

Räisänen, P., Makkonen, R., Kirkevåg, A., and Debernard, J. B.: Effects of snow grain shape on climate simulations: sensitivity tests with the Norwegian Earth System Model, The Cryosphere, 11, 2919–2942, https://doi.org/10.5194/tc-11-2919-2017, 2017.